# The Reaction of the Yeast *Saccharomyces cerevisiae* to Contamination of the Medium with Aflatoxins B_2_ and G_1_, Ochratoxin A and Zearalenone in Aerobic Cultures

**DOI:** 10.3390/ijms242216401

**Published:** 2023-11-16

**Authors:** Grzegorz Kłosowski, Beata Koim-Puchowska, Joanna Dróżdż-Afelt, Dawid Mikulski

**Affiliations:** Department of Biotechnology, Faculty of Biological Sciences, Kazimierz Wielki University, ul. K. J. Poniatowskiego 12, 85-671 Bydgoszcz, Polandjdrozdz@ukw.edu.pl (J.D.-A.); dawidmikulski@ukw.edu.pl (D.M.)

**Keywords:** *Saccharomyces cerevisiae*, aerobic cultures, mycotoxins, toxic stress, oxidative stress, glutathione S-transferase, glutathione peroxidase, heat shock proteins

## Abstract

The mechanisms by which yeast cells respond to environmental stress include the production of heat shock proteins (HSPs) and the reduction of oxidative stress. The response of yeast exposed to aflatoxins B_2_+G_1_ (AFB_2_+G_1_), ochratoxin A (OTA), and zearalenone (ZEA) in aerobic conditions was studied. After 72 h of yeast cultivation in media contaminated with mycotoxins, the growth of yeast biomass, the level of malondialdehyde, and the activity of superoxide dismutase, glutathione S-transferase and glutathione peroxidase were examined; the expression profile of the following heat shock proteins was also determined: HSP31, HSP40, HSP60, HSP70, and HSP104. It was demonstrated that at the tested concentrations, both AFB_2_+G_1_ and ZEA inhibited yeast biomass growth. OTA at a concentration of 8.4 [µg/L] raised the MDA level. Intensified lipoperoxidation and increased activity of SOD and GPx were observed, regardless of the level of contamination with ZEA (300 µg/L or 900 µg/L). Increased contamination with AFB_2_+G_1_ and OTA caused an increase in the production of most HSPs tested (HSP31, HSP40, HSP70, HSP104). ZEA contamination in the used concentration ranges reduced the production of HSP31. The response of yeast cells to the presence of mycotoxin as a stressor resulted in the expression of certain HSPs, but the response was not systematic, which was manifested in different profiles of protein expression depending on the mycotoxin used. The tested mycotoxins influenced the induction of oxidative stress in yeast cells to varying degrees, which resulted in the activation of mainly SOD without GST mobilization or with a small involvement of GPx.

## 1. Introduction

The yeast *Saccharomyces cerevisiae* has been used in industrial processes for many centuries. Understanding the metabolic properties of yeast allowed for effective use of these microorganisms in many industries, including bakery, production of food, alcohols, and biofuels [1,2,3]. During biomass production, and also when used in industrial fermentation processes, yeast cells are exposed to various types of stressors, including osmotic, temperature, ethanol, acid, oxidative, or toxic stress, caused by various compounds present in the used raw material [2,4]. The numerous environmental stressors to which yeast cells are exposed in industrial processes also include mycotoxins synthesized by filamentous fungi, mainly of the genera *Aspergillus*, *Penicillium*, and *Fusarium* [5]. Starch raw materials, especially cereal grains and semi-finished products and final products obtained from them, may contain toxins such as aflatoxins, OTA, and ZEA [6]. Aflatoxins are produced mainly by *Aspergillus flavus* and *A. parasiticus*, OTA is the result of the metabolic activity of *Penicillium* and *Aspergillus fungi*, and ZEA is produced by *Fusarium* [7,8]. All mycotoxins used in this study are typical contaminants found in corn and wheat grains [9]. Aflatoxin is also found in peanuts, OTA in barley grains and, to some extent, coffee, cocoa, and soybeans; ZEA is present in barley grains, sorghum, and in fruits [7,8,9]. Mycotoxins are considered substances with a broad spectrum of toxicity when present in human food and animal feed [10]. Many countries and regions have set maximum limits for the content of mycotoxins in raw materials, feed, and food products [9]. Aflatoxin B_1_ (AFB_1_) is highly carcinogenic to humans, while OTA has been recognized as a possible carcinogen by the International Agency for Research on Cancer. Aflatoxins are primarily hepatotoxic, while OTA is nephrotoxic [8]. ZEA exhibits estrogenic activity, which results in reproductive disorders or physical changes in the reproductive organs of farm animals [11,12]. Moreover, the removal of secondary mold metabolites from food and feed products is particularly difficult because they are resistant to high and low temperatures, even after long storage [5].

Exposing yeast to stress may result in damage to cell walls and membranes, aggregation of denatured proteins or exposure to free radicals, which may ultimately lead to cell death. There are two main types of responses to environmental stress in yeast: the oxidative stress response (OSR) and the heat shock response (HSR), which uses the cooperation of HSPs to eliminate proteins denatured by the stressor [4]. It has been shown that exposure to aflatoxins, OTA and ZEA may have a destructive effect on the redox potential of cells by stimulating the production of reactive oxygen species [13]. The result of this process is oxidative damage to proteins, DNA, and lipids [8,14]. However, yeast cells do not remain passive against oxidative stress because they have developed effective defense mechanisms. In response to excess reactive oxygen species, two simultaneous enzymatic strategies are activated [15]. The first one is related to the neutralization of reactive oxygen species. It involves superoxide dismutases (SOD), which catalyze the dismutation of superoxide anion radical, and catalases, which neutralize hydrogen peroxide [16]. The second strategy is based on maintaining the redox balance of the cell. Peroxidases, e.g., glutathione peroxidases (GPx), reduce peroxides, especially lipid peroxides, using electron donors such as the tripeptide reduced glutathione (GSH). Glutathione transferases, in turn, constitute a link between the two indicated defense strategies, as they can exhibit peroxidase activity and conjugate GSH with exogenous and endogenous electrophilic compounds, including xenobiotics, which allows for the removal and detoxification of GSH-conjugated molecules [15,17].

Another line of defense activated by yeast cells under conditions of environmental stress are heat shock proteins (HSPs). Their task is to process incorrectly folded proteins, prevent the formation of protein inclusions and eliminate irreversibly damaged proteins. In yeast cells exposed to a stressor, proteins become denatured; in turn, misfolded proteins tend to aggregate. Denatured proteins negatively affect the entire cell metabolism, disturbing cell proliferation or the fermentation process [4,18]. Proteins from the HSP family are synthesized not only under stressful conditions, but they constitute up to 10% of the total protein present in the cell under optimal conditions and are then involved in protein translocation, assembly and folding. Due to their functions, they are called “molecular chaperones”. Depending on their size (expressed in kDa), HSPs differ in their role in the response to stress; published reports present analyzes of HSP31, HSP40, HSP60, HSP70, and HSP104 [4,19]. The homodimeric HSP31 protein in *Saccharomyces cerevisiae* is a member of the DJ-1/Pfpl/ThiJ superfamily. Its role is to protect proteins against stress, including oxidative stress. The protein is expressed in the late phases of cell growth and is an important factor supporting survival in nutrient-limited conditions. Studies have shown that in the presence of H_2_O_2_, i.e., under conditions of oxidative stress, HSP expression is increased [18,20]. HSP40 proteins (members of classes I and II) are composed of a J domain, binding the ATPase domain of HSP70, followed by a glycine- and phenylalanine-rich region. The main function of HSP40 is to direct non-native proteins to HSP70. The cooperation of both heat shock proteins includes the processes of translation, translocation, and folding of protein substrates. The role of HSP40 is to stimulate ATP hydrolysis, which supports ADP-bound HSP70 to increase its affinity for unfolded proteins. A probable model of cooperation between HSP40 and HSP70 assumes that HSP40 delivers unfolded proteins to HSP70, where the folding process takes place. HSP40 is able to bind unfolded proteins on its own without the interaction with HSP70, to prevent them from aggregation [21,22].

The chaperone protein HSP60 occurs mainly in mitochondria, protecting proteins imported there from aggregation and denaturation [19,23]. Moreover, its role in binding ssDNA and stimulating nuclear DNA replication in vitro has also been demonstrated [23]. This protein cooperates with HSP10 in the process of refolding proteins after their translocation from the cytoplasm to the mitochondrion [19,24].

In response to a harmful thermal, oxidative, or xenobiotic stress factor, the concentration of HSP70 protein increases. This protein plays a role in cellular processes such as protein folding, assembly, secretion, and degradation. The HSP70 chaperones found in *Saccharomyces cerevisiae* are a multigene family of proteins that include mitochondrial macromolecules synthesized in response to yeast starvation conditions. Their main function is to facilitate the ER-associated degradation of non-native proteins. They are also believed to play a role in cooperation with other proteins involved in the defense against stress factors [25].

The HSP104 protein found in *Saccharomyces cerevisiae* is a member of the Hsp100/Clp AAA+ ATPase subfamily, with a hexameric, ring structure, which folds aggregated proteins in cooperation with HSP70. HSP104 can extract proteins from aggregates and thread them through its axial channel. After such a translocation, the protein returns to its native conformation and, properly folded, can perform its function [26,27].

Thanks to the effective action of all HSP fractions, yeast cells are able to maintain stability despite the influence of various stressors, ranging from those commonly affecting cells during multiplication or fermentation, to specific ones to which microorganisms are exposed in special conditions, such as increased concentration of mycotoxins in the culture medium.

The yeast *Saccharomyces cerevisiae* is treated as a model organism due to its homology to eukaryotic organisms. For this reason, their enzymatic response to oxidative stress has been widely analyzed in numerous studies [28,29,30]. However, there are still no reports on the possible impact of mycotoxins such as AFB_2_+G_1_, OTA, and ZEA on the activation of the antioxidant system of *Saccharomyces cerevisiae* and the production of heat shock proteins in these microorganisms.

For this reason, the aim of the study is to investigate the metabolic response of the yeast *S. cerevisiae* to stress caused by the presence of mycotoxins in the medium, i.e., AFB_2_+G_1_, OTA and ZEA. The current literature review shows that these factors have not yet been fully characterized as stressors in processes involving *S. cerevisiae*, as researchers’ attention has mainly focused on the reduction of yeast viability and fermentation activity due to the presence of these stressors in the fermentation medium. The presented results add to the existing knowledge and can be helpful in better understanding the adaptive mechanisms of microorganisms that help them survive in the presence of stress factors.

## 2. Results

### 2.1. Biomass

The results shown in Figure 1 clearly indicate the influence of mycotoxins, within the range of applied contaminant levels, on the growth of the biomass of the yeast *S. cerevisiae* under aerobic conditions. Both in the case of aflatoxins B_2_+G_1_ and ZEA contamination, a statistically significant (*p* < 0.05) higher average biomass concentration (mg/mL) was found in control samples (10.57 ± 0.71) compared to that observed in samples of media contaminated with mycotoxins. The contamination of the medium with these mycotoxins, quantitatively corresponding to their average maximum concentrations reported in the literature, resulted in a decrease in cell biomass to 8.72 ± 0.145 and 7.08 ± 0.195 mg/mL, respectively; when the contamination level was higher, biomass decreased to 8.75 ± 0.468 and 8.66 ± 1.00 mg/mL, respectively. A higher level of OTA contamination (8.4 µg/L) resulted in a statistically lower biomass concentration of 9.04 ± 0.77 mg/mL compared to that observed in media with a lower level of contamination (2.8 µg/L), where the biomass content was 11.86 ± 0.40 mg/mL. However, it should be noted that no statistically significant differences in *S. cerevisiae* biomass production were found between the control sample and OTA contaminated samples.

### 2.2. The Influence of Mycotoxins on the Generation of Oxidative Stress in the Yeast Saccharomyces Cerevisiae

#### 2.2.1. Aflatoxins B_2_+G_1_

A statistically significant lower degree of lipid peroxidation was found after using a higher dose of AFB_2_+G_1_ (36 µg/L) compared to the control sample. There were no differences between the MDA concentrations in the control sample and in the sample exposed to a lower dose of mycotoxin (12 µg/L). However, superoxide dismutase activity in the sample with lower mycotoxin concentration was significantly higher than in the control sample. No differences were found in GPx and GST activity in samples after AFB_2_+G_1_ application (Figure 2A–D).

#### 2.2.2. Ochratoxin A

In the case of media contaminated with OTA, a higher level of MDA was found in samples exposed to a higher concentration of the mycotoxin (8.4 µg/L) compared to other samples, which indicates that this compound increased oxidative stress in yeast cells. In turn, SOD activity was higher in the sample with a lower OTA concentration (2.8 µg/L) compared to the control and samples with a higher degree of contamination, suggesting that in response to OTA this enzyme was activated, enabling the reduction of oxidative stress in cells. The tested level of OTA contamination had no effect on GPx and GST activity in yeast (Figure 2A–D).

#### 2.2.3. Zearalenone

The analysis of the results confirmed increased lipoperoxidation and activity of SOD and GPx in the sample exposed to a lower concentration of ZEA (300 µg/L), compared to the control sample and the sample containing a higher concentration of the mycotoxin, i.e., 900 µg/L. Moreover, a higher concentration of MDA was found in the sample with a higher concentration of ZEA than in the culture without the mycotoxin. Contamination of the media with ZEA had no effect on GST activity (Figure 2A–D).

### 2.3. Production of HSPs in Response to Mycotoxins

#### 2.3.1. Aflatoxins B_2_+G_1_

As shown in Figure 3A, Appendix A, the production of HSP31 was three times higher in *S. cerevisiae* cultures on media with a higher degree of contamination with aflatoxins B_2_+G_1_ (36 µg/L). These samples also showed an increased average level of production of HSP40, HSP70, and HSP104, higher than in the control samples by 32%, 27%, and 38%, respectively. Significant production of HSP60 and HSP70, approximately twice as high as in the control sample, was found in cultures on media with lower concentrations of these mycotoxins (12 µg/L). Production levels of HSP31 and HSP104 in these samples were the same or lower than those in the corresponding controls (Appendix A). As shown in Figure 2B, Appendix A, a significant reduction in the production of HSP40 was found compared to uncontaminated cultures and those with a higher aflatoxin B_2_+G_1_ contamination.

#### 2.3.2. Ochratoxin A

Data presented in Figure 3A,B,D,E clearly indicate an increased level of production of HSP31, HSP40, HSP70, HSP104 proteins by *S. cerevisiae* in response to higher OTA contamination (8.4 µg/L). Particularly high production, approximately three times higher than in the control, was observed for HSP31 (Figure 3A, Appendix A). The average level of HSP104 was also found to be significantly increased, by approximately 60% compared to control samples, in cultures with a lower degree of OTA contamination (2.8 µg/L) (Figure 3E). In turn, in samples containing a lower dose of OTA, HSP31, HSP40, HSP70 proteins were produced at lower levels (27%, 32%, 17%, respectively) compared to the control sample.

#### 2.3.3. Zearalenone

Increased production of HSP40 proteins by approximately 50% compared to the control (Figure 3B, Appendix A) was found in *S. cerevisiae* cultures with a higher degree of ZEA contamination (900 µg/L). Both lower and higher doses of mycotoxins resulted in increased production of HSP70 compared to uncontaminated control cultures. The results also indicate increased production of HSP104 in samples exposed to lower doses of mycotoxin, i.e., 300 µg/L (Figure 3E). However, after applying both lower and higher doses of ZEA, reduced production of HSP31 proteins was observed, on average by 28% and 65%, respectively; the production of HSP60 proteins decreased by approximately 12%. The production of HSP40 in the lower dose samples was also reduced compared to the control group by approximately 38% (Appendix A). The production of HSP104 under the influence of a higher dose of this mycotoxin was only slightly lower (ca. 10%) than the production of these proteins observed in uncontaminated cultures.

## 3. Discussion

The results obtained in this study indicate a reduction in the growth of yeast biomass in aerobic cultures on media contaminated with mycotoxins or the lack of this effect in the case of OTA compared to uncontaminated control samples. Also [31] confirmed the influence of mycotoxins, including ZEA, on the propagation of the yeast *S. cerevisiae* lager and ale strains. They also showed that the sensitivity of these two yeast strains was determined by the temperature, culture method, and the dose of mycotoxin. Researchers found no reduction in *S. cerevisiae* biomass after treatment with AFB_1_, OTA and ZEA at both higher doses (5 μg/mL, 4 μg/mL, 20 μg/mL) and lower doses (2.5 μg/mL, 2 μg/mL, 2 μg/mL), respectively [32]. Dziuba et al. (2008) showed that even relatively high concentrations of OTA, i.e., from 2.5 to 50 µg, had no toxic effect on brewer’s yeast strains [33]. The ZEA contamination had an adverse effect on yeast growth only at the highest doses (50 and 100 µg/mL). It seems likely that differences in the structure of the cell wall and membrane, manifested even within the same species, and the ability to maintain their integrity, may affect the penetration of mycotoxins into the cell, determining the degree of sensitivity of cells to them, which in turn may affect the activation a number of cellular response processes [32,33]. In the context of the above studies and the results of this study, it can be concluded that the type of mycotoxin, its concentration in the medium and probably also the exposure time and culture conditions influence the growth of yeast cell biomass. In the case of the cumulative effect of AFB_2_+G_1_ on yeast cells, there was a noticeable increase in the production of almost all HSPs (except HSP60) at higher doses of these mycotoxins (36 µg/L) applied in aerobic conditions. HSP31 expression was more than three times higher in samples with higher concentrations of toxins compared to control samples. Reports on HSP31 in *S. cerevisiae* provided some information regarding the role of this protein in the response to oxidative stress. Mycotoxins from the aflatoxin group are involved in generating oxidative stress, which is based on free radical mechanisms [34]. However, based on the data obtained in this study regarding the concentration of MDA as an indicator of oxidative stress, a different mechanism can be suggested. The observed lower lipid peroxidation when yeast cells were exposed to higher concentrations of mycotoxins challenges the hypothesis that oxidative stress is the main mechanism of the toxic effect of aflatoxins on yeast cells. It should be noted that apart from the reported increase in superoxide dismutase activity in aerobic cultures exposed to a lower dose of aflatoxin B_2_+G_1_, no activation of other components of the antioxidant system was observed. Also, a noticeable increase in the production of other proteins, i.e., HSP40, HSP70, and HSP104 after exposure to a higher dose of aflatoxins, indicates an important role of heat shock proteins in the defense response of yeast to this type of stressor. It can therefore be reasonably assumed that due to the lack of induction of antioxidant enzymes or increase in MDA concentration, toxic stress caused by aflatoxins is not associated with free radical reactions [35]. When lower doses of aflatoxins B_2_+G_1_ were used, a significant increase in the expression of HSP60 and HSP70 proteins was observed, while in yeast exposed to higher concentrations of mycotoxins, the increase in the production of these HSPs was small or the values were even lower than in the control, which may suggest the inhibition of protein expression at higher doses of mycotoxins or activation of alternative cell defense mechanisms. The results of this study can only be compared with reports on other types of cells and other mycotoxins, because the authors did not find any published reports similar to those presented in this study that would focus on the impact of aflatoxins on the expression of HSPs in *S. cerevisiae*. Few authors reported an increase in the expression of proteins from the HSP70 family when cells, e.g., porcine alveolar macrophages, were exposed to aflatoxins, while others described an unclear mechanism of action of these proteins when using other mycotoxins, i.e., OTA [36,37].

In aerobic cultures of *S. cerevisiae* treated with OTA, similarly to cultures with aflatoxins, an increase in the expression of most of the analyzed proteins (HSP31, HSP40, HSP70, HSP104) was found when cells were exposed to higher concentrations of this mycotoxin (42 µL/100 mL of medium). In the case of HSP31, the increase compared to the control was threefold. High increases were also observed for HSP40 (approximately 50% higher than in the control sample). The response in terms of HSP104 protein production to both OTA concentrations used (2.8 µg/L and 8.4 µg/L of medium) was similar, and the increase in expression was over 50% compared to the culture in mycotoxin-free medium. The response associated with HSP70 when exposed to higher concentrations of OTA was different than in the case of aflatoxins, as an expected increase in the expression of this protein was observed, which, in combination with the higher MDA concentration, may support the theory of oxidative damage as the mechanism of the toxicity of this compound [38]. Based on the results obtained, it can be concluded that, also in the case of yeast cells, this mycotoxin may have an inhibitory effect on the activity of superoxide dismutase in cultures exposed to higher doses of mycotoxins. However, the influence of OTA on cells is still not entirely clear. There is no data on the effect of this compound on the production of HSPs or the activation of the antioxidant system, including antioxidant enzymes in yeast cells. Reports on other cell types (Hep G2 and Vero) indicate that the HSP70 response is not systematic for OTA-induced cellular stress [37]. The divergent results may indicate the complexity of HSP protein induction, which encourages further studies, perhaps including even higher doses of mycotoxin, because the observed inhibition of yeast biomass growth with the doses of OTA used was not significantly high. The lack of activation of GPx and GST under oxidative stress conditions may also indicate the mobilization of other components of antioxidant mechanisms, e.g., untested catalase, whose task is to detoxify hydrogen peroxide formed as a result of dismutation of superoxide anion radical.

The third of the tested mycotoxins, ZEA, caused a decrease in HPS31 production at both the lower and higher concentrations of the mycotoxin compared to the control sample. A similar result was obtained with HSP60. It can therefore be assumed that the observed inhibition of the production of these proteins may be the result of the mycotoxins inhibiting the factors necessary for the production of this protein fraction or substituting other cellular response mechanisms. A clear increase in HSP40 expression (by over 50%) observed at a higher concentration of ZEA in the culture medium may indicate the importance of this protein in the defense response against the applied stressor. Interestingly, HSP70 and HSP104 were overexpressed at lower concentrations of ZEA, while at higher concentrations this effect was no longer observed (the values did not differ significantly from the control values), Hassen et al. (2005), when examining the response of Hep G2 cells to ZEA contamination, noticed an increase in the production of HSP70 even at non-toxic doses, which, according to the authors, was supposed to be an indicator of toxicity [39]. The next study by these authors [35] showed an increase in the production of HSP70 in the same cells, also at non-toxic concentrations, combined with the formation of oxidative damage. It was therefore concluded that this mycotoxin caused oxidative stress in Hep G2 cells. The results of our research only partially support this hypothesis. For lower ZEA concentrations, we observed high expression of HSP70 and HSP104, increased levels of lipid peroxidation, but also high concentrations of superoxide dismutase, which indicated oxidative stress. The results obtained for higher concentrations of the mycotoxins used are difficult to interpret (similar or even lower production of HSP70 and HSP104 compared to the control; low SOD value), but the increased level of lipid peroxidation still indicates oxidative stress. Mike et al. 2013 showed that the use of ZEA at a dose of 500 μmol/L on the fission yeast *Schizosaccharomyces pombe* resulted in a 66% decrease in glutathione concentration, the accumulation of superoxide anion radical and hydrogen peroxide, a decrease in sterol concentration, and the fragmentation of the cell nucleus [40]. As a result of the increase in the concentration of reactive oxygen species, adaptive mechanisms were initiated by activation of the Pap1 transcription factor and, thus, an increase in the activity of antioxidant enzymes, including superoxide dismutase, catalase, glutathione reductase, and glutathione S-transferase, associated with a decrease in the activity of glutathione peroxidase. Therefore, based on our results and the cited reports, it can be concluded that ZEA toxicity may be related to the generation of oxidative stress in the cell and, consequently, also to the mobilization of components of the antioxidant response. However, it is likely that many combinations of variables, such as the sensitivity of the yeast species to ZEA, the dose of mycotoxins, or the culture conditions, could result in a different pattern of defense response.

Our results showing the responses of *S. cerevisiae* to stress caused by the presence of mycotoxins in the culture medium are consistent with some studies emphasizing the lack of a clear cellular response to stress conditions caused by the presence of mycotoxins in the culture medium [36,37]. On the other hand, the results of other studies should be taken into account, which indicate, e.g., the protective role of HSPs in stress caused by various factors [35]. For example, studies on the response of *S. cerevisiae* to other toxic stressors, such as by-products of lignocellulose pretreatment, showed an increase in the expression of HSP60 in the presence of furan aldehydes [41]. To sum up, the response of yeast cells to the presence of a stressor (mycotoxin) in the medium results in the expression of some HSPs, but it is not systematic, and, depending on the mycotoxin used, a different expression profile of heat shock proteins appears. The tested mycotoxins have different effects on the induction of oxidative stress in yeast cells, which translates mainly into the activation of superoxide dismutase, without GST mobilization, with a GST involvement only in the case of ZEA GPx. The presented research results constitute a basis for further investigations, using other toxins, other doses or expanding the profile of the tested HSPs and the number of analyzed oxidative stress indicators.

## 4. Materials and Methods

### 4.1. Saccharomyces Cerevisiae Ethanol Red Strain

The commercial yeast strain Ethanol Red (Lesaffre Advanced Fermentations, Marcq-en-Baroeul, France), commonly used in the spirit industry and in bioethanol production in the form of dried preparation, was used in the study. Yeast milk for inoculation of the culture medium was prepared by mixing 1 g of the product with 10 mL of 0.9% NaCl (21 °C, stirring on a magnetic stirrer for 25 min, sterile conditions).

### 4.2. Materials

All mycotoxins, lysis buffer components, Bovine Serum Albumin were purchased from Sigma-Aldrich (St. Louis, MO, USA). The mycotoxins used in the study were added to the culture as ethanol solutions; their concentrations are given in Table 1. Aflatoxins B_2_ and G_1_ were mixed in one solution due to the simultaneous contamination of starch raw materials with these mycotoxins observed in nature [42]. Reagents for MDA assay were purchased from Merck (Darmstadt, Germany). Antioxidant enzyme activity assay kits were purchased from Cayman Chemical Company (Ann Arbor, MI, USA). Protein size standard and secondary antibodies were provided by Bio-Rad (Hercules, CA, USA). Other antibodies (primary and secondary) were purchased from Abcam (Cambridge, UK), Jackson ImmunoResearch Lab. Inc. (Ely, Cambridgeshire, UK) and OriGene (Rockville, MD, USA).

### 4.3. Microbiological Cultures

Ethanol Red yeast cultures were carried out in triplicate for each of the two levels of contamination of the culture medium with each mycotoxin (Table 1). The level of media contamination was determined based on the literature data reporting the real level of mycotoxin contamination of starch raw materials, including corn grains [42,43,44]. Concentrations three times higher than the actual (lower) doses were also applied and were referred to as higher doses.

The culture was carried out in aerobic conditions, in baffled culture flasks with membrane caps, in 100 mL of PYN liquid medium according to [45], containing 20 g/L glucose, pH = 5.6. After inoculation with 200 µL of yeast milk, cultures were stirred (70 rpm) in a water bath at 28 °C for 72 h. Ethanol solutions of mycotoxins were added in appropriate doses (Table 1) after sterilization of the culture medium and before inoculation with microorganisms. An equal volume of ethanol was introduced into each medium, including the controls, to eliminate its possible influence on the culture conditions and experimental results.

### 4.4. Determination of Yeast Biomass Concentration

The concentration of yeast biomass was determined using a spectrophotometer (Spectroquant^®^ Pharo 300, Merck, Darmstadt, Germany) after 72 h of yeast cultivation in aerobic conditions. Samples of the culture medium (5 mL) were collected and centrifuged twice (10,000 rpm, 10 min, 20 °C), then the cells were rinsed with 0.9% NaCl. The yeast biomass was suspended in isotonic saline, and then the optical density (OD_500_) was determined; the results are expressed in mg/mL.

### 4.5. Lysis of Yeast Cells

#### 4.5.1. Lysis of Yeast Cells to Assess the Level of Oxidative Stress

To analyze the MDA concentration, a 1 mL sample of cell biomass collected from the culture was centrifuged (3500× *g*, 4 min, 4 °C) and rinsed three times with 0.1% trichloroacetic acid. The resulting cell pellet was homogenized using glass beads; after centrifugation, the supernatant was used for further analyses. In order to obtain cell lysate for the assessment of superoxide dismutase activity, 1 mL of yeast cell biomass was centrifuged (1000× *g*, 10 min, 4 °C) and homogenized using glass beads in chilled 20 mM HEPES buffer, pH = 7.2, containing 1 mM EGTA, 210 mM mannitol and 70 mM sucrose. The suspension thus obtained was centrifuged (1500× *g*, 5 min, 4 °C), the supernatant was collected and stored at –80 °C until further analyses. A 1 mL culture sample taken for the purpose of lysing yeast cells and analyzing the activity of glutathione S-transferase was centrifuged (1000× *g*, 10 min, 4 °C), then the pellet was homogenized using glass beads in a chilled buffer containing 100 mM potassium phosphate at pH 7.0 and 2 mM EDTA. The lysate was centrifuged (10,000× *g*, 15 min, 4 °C), and the obtained supernatant was stored at –80 °C until further analyses. The cell lysis procedure for assessing glutathione peroxidase activity was similar to that for glutathione S-transferase, but a buffer containing 50 mM Tris-HCl, pH = 7.5, 5 mM EDTA, and 1 mM DTT was used for homogenization. The supernatant was stored at –80 °C until further analyses. Three repetitions of the above procedures were performed for each flask with culture medium.

#### 4.5.2. Lysis of Yeast Cells to Determine the Spectrum of Heat Shock Proteins

At the beginning of the cell lysis procedure, the yeast biomass was separated from the culture medium. From each culture, 40 mL samples were taken and centrifuged (4500 rpm, 5 min, 4 °C), then the pellets were suspended in 2 mL of distilled water at 4 °C, vortexed, and 200 µL aliquots were transferred to Eppendorf tubes. The solution was centrifuged (10,000 rpm, 10 min, 4 °C) and the resulting pellet was frozen for 5 min at –80 °C. The pellet was then suspended in 400 µL of 0.1 M NaOH and, after 5 min of incubation, the samples were centrifuged (10,000 rpm, 5 min, 21 °C), and the pellet was dissolved in 200 µL of lysis buffer containing 4% SDS, 20% glycerol, 120 mM Tris-HCl (pH = 6.8), and 10% betamercaptoethanol. The samples were incubated in a water bath for 8 min at 80 °C, and after centrifugation, the supernatant was used to determine the total protein concentration using the Bradford method [46]. Protein concentration is presented in mg/mL.

### 4.6. Determining the Level of Oxidative Stress

#### 4.6.1. Analysis of Malonaldehyde (MDA) Concentration

MDA concentration was determined using the methodology of [47] modified by [48]. A reference sample was prepared by mixing 200 µL of H_2_O, 20 µL of 2% butylhydroxytoluene (BHT) in ethanol, 1 mL of a 15% solution of trichloroacetic acid (TCA) in 0.25 M HCl, and 1 mL of 0.37% thiobarbituric acid (TBA) in 0.25 M HCl. Two test tubes were prepared for each analyzed sample. One of them was filled with 200 µL of the lysed sample, 20 µL of BHT in ethanol, 1 mL of a 15% TCA in HCl solution, and 1 mL of 0.37% TBA in HCl. Instead of TBA in HCl solution, 1 mL of distilled water was introduced into the second test tube. Then the samples were vortexed and heated in a water bath for 10 min at 100 °C. After cooling, the samples were centrifuged and the supernatant was used to measure absorbance at λ = 535 nm. The concentration of MDA in the samples was calculated using the millimolar absorbance coefficient (156 mmol^−1^cm^−1^) and expressed in µM/mg of biomass.

#### 4.6.2. Analysis of Superoxide Dismutase (SOD) Activity

Superoxide dismutase activity was determined using the standardized Superoxide Dismutase Assay Kit (Cayman Chemical Co., Item No. 706002, Ann Arbor, MI, USA) according to the methodology outlined in the manual. The analysis was performed in 96-well plates using a Synergy HTX multi-mode reader, Biotek, Santa Clara, CA, USA. First, SOD solutions with activities ranging from 0 to 0.25 U/mL were prepared and applied to the plate in duplicate, then 10 µL of the analyzed samples were added to the remaining wells, and finally 200 µL of Radical Detector solution (tetrazolium salt solution) was added to all wells. The reaction was started by adding 20 μL of xanthine oxidase solution to each well, and then the plate was shaken for 20 min at room temperature. The absorbance was measured at λ = 450 nm. Enzymatic activity was calculated based on the SOD standard curve and expressed in U/mL and then in U/mg of biomass.

#### 4.6.3. Analysis of Glutathione S-Transferase (GST) Activity

Glutathione S-transferase activity in the cell lysate was determined using the standardized Glutathione S-transferase Assay Kit (Cayman Chemical Co., Item No. 703302, Ann Arbor, MI, USA). The analysis was performed in 96-well plates using a Synergy HTX multi-mode reader, Biotek, Santa Clara, CA, USA. Nonenzymatic background samples, prepared by adding 170 μL of Assay Buffer solution (100 mM potassium phosphate, pH 6.5, containing 0.1 Triton X-100) and 20 μL of glutathione, were applied to three wells of the plate. Positive control samples were placed in three wells by mixing 150 µL of Assay Buffer, 20 µL of glutathione and 20 µL of control GST. The remaining wells were filled with 150 µL of Assay Buffer, 20 µL of glutathione and 20 µL of analyzed samples. The reaction was started by adding 10 µL of CDNB (ethanol solution of 1-chloro-2,4-dinitrobenzene) to all wells, then the plate was shaken for 10 s and the absorbance was measured at λ = 340 nm every 1 min at 5 time points. Enzymatic activity was calculated from the difference in absorbance per minute, taking as reference the results obtained for a non-enzymatic background. The activity of glutathione S-transferase was expressed in nmol/min/mL, and, after taking into account the biomass concentration in the sample, in nmol/min/mg.

#### 4.6.4. Analysis of Glutathione Peroxidase (GPx) Activity

The activity of glutathione peroxidase in the samples was determined using the standardized Glutathione Peroxidase Assay Kit (Cayman Chemical Co., Item No. 703102, Ann Arbor, MI, USA). The analysis was performed in 96-well plates using a Synergy HTX multi-mode reader, Biotek, Santa Clara, CA, USA. Three reference samples were prepared by filling three wells with 70 µL of Assay Buffer (50 mM Tris-HCl, pH 7.6, with 5 mM EDTA), 50 µL of GPx-Co-Substrate Mixture (solution of lyophilized glutathione and glutathione reductase) and 50 µL of NADPH. The next three wells were filled with 50 µL of Assay Buffer, 50 µL of GPx-Co-Substrate Mixture, 50 µL of NADPH, and 20 µL of diluted GPx. The remaining wells were filled with 50 µL of Assay Buffer, 50 µL of GPx-Co-Substrate Mixture, 50 µL of NADPH, and 20 µL of lysed samples. The reaction was started by adding 20 µL of cumene hydroxide to each well. The plate was shaken for 10 s and the absorbance was measured at λ = 340 nm every minute at 5 time points. Glutathione peroxidase activity was calculated from the difference in absorbance per minute, taking as reference the results obtained for a non-enzymatic background. The activity was expressed in nmol/min/mL, and, after taking into account the biomass concentration in the sample, in nmol/min/mg.

### 4.7. Determination of Heat Shock Proteins

Heat shock proteins HSP31, HSP40, HSP60, HSP70 and HSP104 were determined using SDS-PAGE electrophoretic separation, semi-dry transfer of proteins to the membrane and chemiluminescence visualization using the Chemi-Doc MT Imager, Bio-Rad. The first step was the electrophoretic separation of the samples (10 µg per well) and the protein size marker (Precision Protein Standards 10–250 kD, BioRad, Hercules, CA, USA; 5 µL per well) for 1 h, at 110 V. Ready-to-use stain-free electrophoresis gels, 4–15% MP TGX Stain Free, BioRad, Hercules, CA, USA, were used for the protein separation with the ChemiDoc MT Imager device, which enabled the visualization of proteins without staining them. Semi-dry transfer of proteins onto an LF PVDF membrane was performed using Trans-Blot Turbo RTA Mini LF PVDF, BioRad, Hercules, CA, USA, with the Trans-Blot Turbo Transfer System, BioRad, Hercules, CA, USA, according to the TURBO “1 Mini PROTEAN TGX” protocol. The first step was to equilibrate the transfer stacks in the transfer buffer and LF PVDF membranes, first in methanol and then in the buffer. The assembled transfer sandwiches with gels placed on the membranes were transferred to the Trans-Blot Turbo device, where the transfer lasted 3 min at 1.3 A. The membranes were then visualized using the stain-free blot protocol on a Chemi-Doc MT Imager. Subsequently, the membranes were blocked with 2% Bovine Serum Albumin overnight at 4 °C, and then rinsed for 10 min in TBST (1xTBS, 0.5 mL/L Tween20). Then primary antibodies were added (HSP31 Goat Ig Cat No: R34015-1004G, nsj Bioreagents, San Diego, CA, USA; HSP 40 Ms mAb to HSP40 ab 74442, HSP 70 Ms mAb to HSP70 ab 2787, HSP 104 Rb pAb to HSP 104 ab 2924, Abcam, Cambrige, UK; HSP 60 Host: Mouse TA 326374, OriGene, Rockville, MD, USA) dissolved in 2% Bovine Serum Albumin; the membranes were then incubated for 60 min at room temperature. After washing the membranes 6 times for 5 min in TBST, secondary antibodies were added (Anti-Goat IgG (H+L) Code: 305-035-003, Jackson ImmunoResearch Lab. Inc., Ely, Cambridgeshire, UK; Goat Anti -Mouse IgG (H+L)- HRP Conjugate Cat# 170-6515; Goat Anti-Rabbit IgG (H+L)- HRP ConjugateCat# 170-6515, Bio-Rad, Hercules, CA, USA) dissolved in 2% albumin. Unbound antibodies were removed by rinsing the membranes 6 times for 5 min in TBST, then HRP substrate for chemiluminescence was applied to each of the five membranes incubated with a given antibody (1:1) and, after a short incubation, chemiluminescence was analyzed with a Trans-Blot Turbo device according to the Chemiluminescence protocol. Transfer membranes and chemiluminescent HSP membranes were analyzed using Image Lab 6.1 software, Bio-Rad, Hercules, CA, USA. This program allowed for normalization of the amount of proteins added to wells, automatic detection of separated proteins, and generation of semi-quantitative reports comparing chemiluminescence-detected proteins with controls.

### 4.8. Statistical Analysis

Statistica 13.1 and Excel version 2301 were used for the statistical analysis of the results. The Shapiro–Wilk test was used to check the normality of data distribution in groups. Main effects ANOVA test and then Tukey’s HSD test were used to analyze the impact of two different doses of mycotoxins on the dependent variables, i.e., the concentration of malonaldehyde (MDA), the activity of enzymes: superoxide dismutase (SOD), glutathione S-transferase (GST), and glutathione peroxidase (GPx). Statistical analysis was performed at a significance level of 0.05.

## 5. Conclusions

Interactions between yeast and mycotoxins present in raw materials that are used to prepare media are the subject of research by many scientists due to the wide use of yeast and the common problem of mycotoxin contamination of various raw materials and food products. However, the mechanism of mycotoxin toxicity to yeast cells, as well as their reaction to the presence of these toxic compounds, are still not sufficiently understood. The results of this study show that, contrary to many available reports suggesting high tolerance of *S. cerevisiae* to this type of toxic stressors, the presence of mycotoxins may negatively affect yeast aerobic cultures. In particular, aflatoxins B_2_+G_1_ and ZEA can inhibit the growth of yeast biomass despite the activation of cell defense mechanisms. Both OTA and ZEA at higher doses induced oxidative stress, which was associated with increased lipid peroxidation and elevated SOD activity; lower doses of ZEA also increased GPx. In response to the toxic effects of higher concentrations of aflatoxin B_2_+G_1_ and OTA, yeast cells significantly increased the production of HSP31 and, to a lesser extent, HSP40, HSP70, HSP104. In turn, exposure to ZEA at a higher dose resulted in an increase in the production of HSP40 and HSP70 but also a significant suppression of HSP31 and HSP60. It was demonstrated that as a result of exposure to higher concentrations of mycotoxins, there was an increase in the production of HSP40, while at lower doses of stressors, the production of this protein was inhibited compared to control cultures. It can therefore be concluded that both the type of mycotoxin and its concentration can stimulate various cellular response mechanisms with a significant degree of complexity. The results of this study may be helpful in the process of selecting and improving production strains of *S. cerevisiae* with the use of adaptive mechanisms.

## Figures and Tables

**Figure 1 ijms-24-16401-f001:**
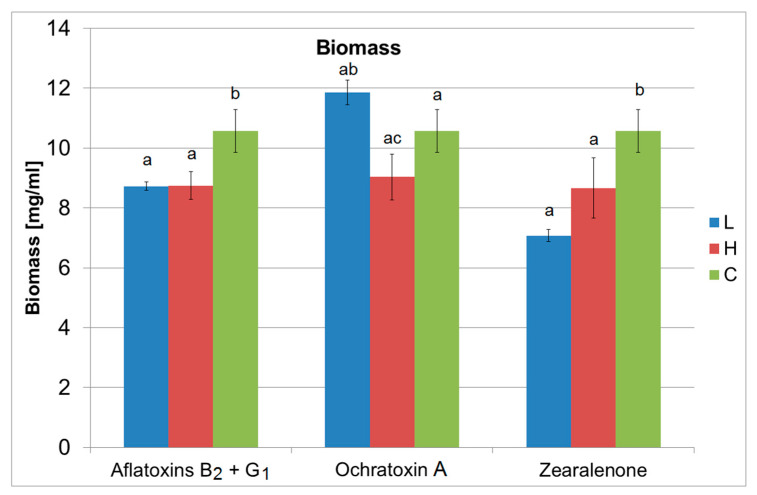
The influence of mycotoxins: aflatoxins B_2_+G_1_, OTA, ZEA on the growth of *Saccharomyces cerevisiae* Ethanol Red biomass after 72 h of culture (L—lower dose of mycotoxins, H—higher dose of mycotoxins, C—control sample; the mean values with different letter index are significantly different (α ≤ 0.05).

**Figure 2 ijms-24-16401-f002:**
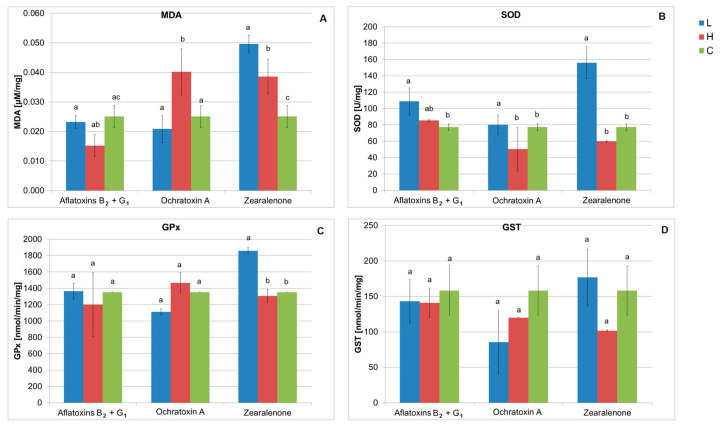
(**A**–**D**) The effect of different concentrations of Aflatoxins B_2_+G_1_, Ochratoxin A, and Zearalenone on the activity of superoxide dismutase (SOD), glutathione peroxidase (GPx), glutathione transferase (GST), and the level of lipoperoxidation (MDA) in *S. cerevisiae* Ethanol Red (L—lower dose of mycotoxins, H—higher dose of mycotoxins, C—control. The mean values with different letter index are significantly different (α ≤ 0.05).

**Figure 3 ijms-24-16401-f003:**
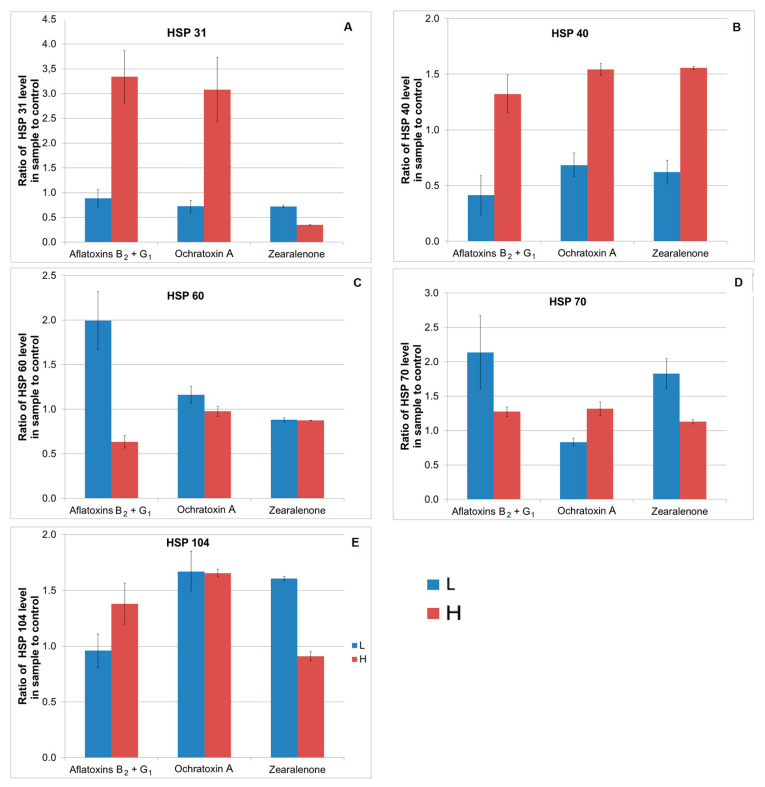
(**A**–**E**) Production of HSP31 (**A**), HSP40 (**B**), HSP60 (**C**), HSP70 (**D**), HSP104 (**E**) by the yeast *S. cerevisiae* Ethanol Red in response to mycotoxins: AFB_2_+AFG_1_, OTA, ZEA. L—lower dose of mycotoxin, H—higher dose of mycotoxin.

**Table 1 ijms-24-16401-t001:** Mycotoxin contamination of culture media.

Mycotoxin	Maximum Concentration Observed in the Raw Material [µg/kg]	Concentration in the Culture Medium [µg/L]	Ethanol Solution Used to Contaminate Media [mg/25 mL]	Concentration in the Culture Medium [µg/L]	
				Lower Concentration (Based on Literature Data)	Higher Concentration
AFB_2_ + AFG_1_	24.2 [43]	12.1 [43]	2	12	36
OTA	5.3 [44]	2.7 [44]	5	2.8	8.4
ZEA	603 [42]	302 [42]	25	300	900

## Data Availability

Data are contained within the article or Appendix A.

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
