# Peer review of "The Reaction of the Yeast Saccharomyces cerevisiae to Contamination of the Medium with Aflatoxins B2 and G1, Ochratoxin A and Zearalenone in Aerobic Cultures"

_ijms, 2023, doi:10.3390/ijms242216401_

Round 1

Reviewer 1 Report

Comments and Suggestions for Authors

The proposed manuscript is focused on an interesting and current problem namely improving methods for the selection and cultivation of Saccharomyces cerevisiae strains used in various industrial productions - food, brewing, biofuel, chemical industry, medicine, etc. The goal is clearly formulated - to study the response of S. cerevisiae against stress induced by the presence of mycotoxins in the medium. The authors investigated two possibilities involved in the protection of yeast cells: the antioxidant defense and the role of stress proteins. Such studies of the S. cerevisiae model are very rare. The experiments employ both conventional and contemporary methods. However, the manuscript needs more attention.

1.  The section Introduction is written very extensively, in places with superfluous details in a textbook style.

2.  For the morphological response to mycotoxins, only the effect on biomass is presented. For a journal like IJMS, more experiments should be included.

3.  To prove the relationship with oxidative stress, more methods need to be used, such as the level of radicals, the amount of damaged proteins, DNA damage, and the activity of catalase and glutathione peroxidase.

4.  Results regarding the generation of oxidative stress and antioxidant defense show a wide variety. In my opinion, this study needs to use more biomarkers (see point 3).

5.  Figure 2 does not have a title on the Y ordinate, which makes it difficult for the reader to perceive the obtained results.

6.  Supplementary figures do not contribute to clarifying the results. They lack all possible designations, there is no legend or description of the results presented.

7.  The section Discussion includes many redundant passages. The first passage (page 9, L 266-277) is a repetition of these presented in the Introduction. I recommend that authors start this section with a discussion of their own data rather than literature descriptions.

8.  The discussion includes explanations for which there is no evidence in the section Results. The authors speculate on the background of literary data.

9.  Page 15, L 621-622 The authors surprise us with the recognition that “Because the data are inconclusive, further research into the mechanisms of mycotoxin toxicity in yeast is necessary’. I recommend they offer convincing results in journals like IJMS.

10.  Inaccurate citations are found in the manuscript.

11.  In the section References there are literary sources with errors in the bibliography.

Author Response

Detailed response to the Reviewers' comments – Manuscript ID: ijms-2671165      

We would like to thank the Editor and the Reviewers for their comments and suggestions. We have revised the manuscript point by point according to the Reviewers’ comments. All the suggested changes are marked in yellow in the revised text and described below.

We hope that the quality and readability of our manuscript has been improved.

Response to Reviewer No. 1 comments:

  1. The section Introduction is written very extensively, in places with superfluous details in a textbook style.

According to the Reviewer's suggestion, the Introduction has been verified and corrections have been made to the revised version of the manuscript.

Sentences removed from the introduction:

Page: 1  lines: 34-39

Page: 2  lines: 45; 52-57; 61; 64-65; 74-75; 76-77

Page: 3  lines: 102-103; 107-109; 111-112; 121-122; 126-127

Page: 3-4  lines: 151-154

  1. For the morphological response to mycotoxins, only the effect on biomass is presented. For a journal like IJMS, more experiments should be included.

Studies on the impact of mycotoxins on microorganisms do not indicate that the response to toxins involves morphological changes in cells. The reported toxic effects include inhibition of biomass growth and impairment of some biochemical processes occurring inside the cell. Many studies have shown that the first line of defense of microorganisms exposed to mycotoxins (in toxic concentrations) is their biosorption on the cell wall surface, which may limit the toxic effect and penetration of these compounds into the cell. The biosorption of mycotoxins by microbial biomass is also considered to be an effective decontamination method. For the above reasons, we did not focus on possible morphological changes in yeast cells, but on molecular aspects of the cellular response to stress, which are still poorly understood. We would like to emphasize that the aim of our study was to analyze the expression of heat shock proteins, the activity of antioxidant enzymes and the intensity of lipid peroxidation in response to the applied stressors. Monitoring biomass propagation is one of the first sources of information about toxins and their impact on yeast reproduction. The Reviewer's suggestion to include more experiments to assess morphological response is interesting, but identifying changes at the level of cell morphology was not the goal of this study.

  1. To prove the relationship with oxidative stress, more methods need to be used, such as the level of radicals, the amount of damaged proteins, DNA damage, and the activity of catalase and glutathione peroxidase.
  2. Results regarding the generation of oxidative stress and antioxidant defense show a wide variety. In my opinion, this study needs to use more biomarkers (see point 3).

As the Reviewer aptly noted, many indicators can be used in the analysis of oxidative stress in yeast, including activity of antioxidant enzymes and the level of protein and DNA damage markers. However, an analysis using all potentially applicable oxidative stress markers would be well beyond the scope of a single publication. As the Reviewer stated at the beginning of the review, similar studies are rare. This is why, when initiating our research on the impact of oxidative stress caused by mycotoxins on yeast, we selected indicators involved in various strategies of the cell's response to excess reactive oxygen species. The analysis of glutathione peroxidase activity, mentioned by the Reviewer, has been included in the study. This enzyme contributes to maintaining the redox balance in the cell by reducing peroxides (including lipid peroxides). Superoxide dismutase (SOD), included in the study, catalyzes the dismutation of the superoxide anion radical and is involved in the process of neutralization of reactive oxygen species. Glutathione S-transferase is a bridge between two defense strategies, as it both acts as a peroxidase and conjugates reduced glutathione (GSH) with exogenous and endogenous electrophiles, including xenobiotics; GSH-conjugated molecules can be removed and detoxified. The level of oxidative damage within yeast cell membranes was also assessed by analyzing the concentration of malondialdehyde, one of the lipoperoxidation products. For these reasons, we believe that the selection of the above-mentioned indicators as markers of oxidative stress is justified and sufficient at the current stage of research to assess the impact of selected mycotoxins on the stimulation of antioxidant mechanisms.

  1. Figure 2 does not have a title on the Y ordinate, which makes it difficult for the reader to perceive the obtained results.

As suggested by the Reviewer, a description of the Y axis has been added to Figure 2.

  1. Supplementary figures do not contribute to clarifying the results. They lack all possible designations, there is no legend or description of the results presented.

Markings and legends (descriptions) of the presented results have been added in the Supplementary materials.

  1. The section Discussion includes many redundant passages. The first passage (page 9, L 266-277) is a repetition of these presented in the Introduction. I recommend that authors start this section with a discussion of their own data rather than literature descriptions.

In accordance with the Reviewer's recommendations, the indicated lines were deleted, and the Discussion section begins with an analysis of the results of this study in the context of literature data.

Sentences removed from the discussion:

Page: 8  lines: 266-277; 283-287

Page: 9  lines: 305-307; 315-319; 324-326

Page: 10  lines: 351-354; 378-379; 391-394

  1. The discussion includes explanations for which there is no evidence in the section Results. The authors speculate on the background of literary data.

Since the Reviewer did not indicate specific fragments of the discussion that raised his/her doubts, it has been difficult for us to precisely comment on this remark. Nevertheless, we have re-analyzed the text, tried to take into account the Reviewer's comments, and introduced some alterations to the manuscript. As stated at the beginning of the review, the topic of our study is rarely addressed, so what the Reviewer considered speculative may have resulted from a difficulty in finding comparative data to discuss our results. As stated in the manuscript, many studies have focused on identifying toxic effects in yeast (manifested by inhibition of biomass growth, reduced fermentation activity, qualitative and quantitative changes in volatile metabolic by-products), or testing the suitability of S. cerevisiae for biodegradation or biosorption. The authors' intention was to expand the available knowledge, mainly of technological nature and importance, by providing some insight into poorly understood aspects of the response at the cellular level.

We hope that the changes introduced will meet the Reviewer's expectations.

  1. Page 15, L 621-622 The authors surprise us with the recognition that “Because the data are inconclusive, further research into the mechanisms of mycotoxin toxicity in yeast is necessary’. I recommend they offer convincing results in journals like IJMS.

Unfortunate wording has been removed.

Sentences removed from the conclusion:

Page:  15  lines: 621-622

  1. Inaccurate citations are found in the manuscript.

The citations include 54 items. Unfortunately, the Reviewer did not indicate which of them were imprecise. The authors checked the correctness of the citations again.

Sentences removed from the references:

Page: 17   lines: 708-715; 718-719; 725-726; 737-738

  1. In the section References there are literary sources with errors in the bibliography.

According to the Reviewer's suggestion, the References section has been verified.

Reviewer 2 Report

Comments and Suggestions for Authors

In this paper Kolosowski et al. demonstrate the response of baker's yeast to several fungal toxins. This is done via profiling the expression of several genes, the activity of certain enzymes and the accumulation of a metabolite. 

While the gathered data is probably useful for the understanding of the modes of action of the tested toxins, it is my feeling that this material not presented very convincingly to be accepted in IJMS. In my mind the material, after improving the quality of the presentation and reducing the length of certain sections, could be published in a more focused and specialized journal, perhaps one focusing on food technology, environmental contaminants etc. The reason for my view is that I do not see any real insight into the basic scientific aspects of the mechanisms of action of the toxins or the yeast reponse to them. 

I also think that the figure presentation can be and should be improved considerably. The authors use histograms  for some figures, while they use tables for others, though the nature of the data does not differ between the figures. In my mind, all of them should be presented as histograms, as a more visual and easily readable format. 

The legends on the histograms are redundant (too many legend panels of the same type) and could contain information on concentrations rather that letter acronyms, which are not intutive and provide no clarity. 

Since the repsonses are normalized to the control (i.e. 1), it's possible that including that bar in the plot is also redundant. All of these comments would help reduce the amount of physical and mind-space that the reader needs to expend to comprehend the material in the paper. 

To my mind, the conclusions of the paper are not very clear, while the discussion is overly long with at least some statements that are too speculative to be conclusions, since no data is presented to support them. Such as the statement in line 317-318 regarding the cell cycle. This statement could have been checked via monitoring of the cell cycle state of the yeast. 

Lines 322-324 These statements could also easily be checked directly by monitoring the effects of antioxidant addition on biomass or marker induction

Lastly, non the described responses have been tested for being relevant for the toxic effects of the toxins. This could be done be searching the literature for whether mutations (deletions) of these proteins increased sensitivity to the toxins, or checking whether this was the case directly. 

Overall, I think that the presented body of work is unsufficiently complete for a publication in IJMS, but, with some improvement, might be acceptable in a more specialized journal. 

Author Response

Detailed response to the Reviewers' comments – Manuscript ID: ijms-2671165

We would like to thank the Editor and the Reviewers for their comments and suggestions. We have revised the manuscript point by point according to the Reviewers’ comments. All the suggested changes are marked in yellow in the revised text and described below.

We hope that the quality and readability of our manuscript has been improved.

Response to Reviewer No. 2 comments:

  1. In this paper Kolosowski et al. demonstrate the response of baker's yeast to several fungal toxins. This is done via profiling the expression of several genes, the activity of certain enzymes and the accumulation of a metabolite.

We would like to correct that the first author's name is Kłosowski, and the object of the research was not baker's yeast, but the Ethanol Red strain selected for the spirit industry and bioethanol production. Additional explanation is provided in Section 4. Materials and methods, 4.1. Saccharomyces cerevisiae strain Ethanol red (line 441). This is important because further research is planned on toxic stress caused by selected mycotoxins under ethanol stress conditions during alcoholic fermentation.

  1. While the gathered data is probably useful for the understanding of the modes of action of the tested toxins, it is my feeling that this material not presented very convincingly to be accepted in IJMS.

Although we treat the Reviewer's comments with due respect and attention, it is difficult to comment substantively on the quoted remarks, as they only express certain critical feelings or assumptions of a very general nature, without indicating any specific objections to the presented results.

  1. In my mind the material, after improving the quality of the presentation and reducing the length of certain sections, could be published in a more focused and specialized journal, perhaps one focusing on food technology, environmental contaminants etc. The reason for my view is that I do not see any real insight into the basic scientific aspects of the mechanisms of action of the toxins or the yeast reponse to them.

The above comments are again general in nature. Nevertheless, in line with the Reviewers' suggestions, the way of presenting the results was improved, the Introduction and Discussion were modified, and their volume was reduced. The Reviewer's suggestion to send the manuscript to another specialized journal is somewhat surprising. We would like to point out that the manuscript was submitted to the Special Issue of "Stress Response Research: Yeast as Models" after the Guest Editors confirmed that the topic of the manuscript would be consistent with the profile of the journal and the special issue. The Reviewer's comment that "I don't see any real insight into the basic science behind how the toxins work or how yeast reacts to them" is too general and therefore it is difficult to engage in a substantive discussion with it. Instead, we believe that the wide range of yeast cellular responses to mycotoxin-induced toxic stress described in the article, including many bioindicators to assess the severity of oxidative stress and HSP expression, clearly contradicts such a statement. All the more so since the other Reviewer points out that this type of research, providing an insight into the phenomenon of stress in industrial yeast cultures, is rarely conducted.

  1. I also think that the figure presentation can be and should be improved considerably. The authors use histograms for some figures, while they use tables for others, though the nature of the data does not differ between the figures. In my mind, all of them should be presented as histograms, as a more visual and easily readable format.

According to the Reviewer's suggestion, the presentation of some of the results was changed from tables to figures. This form of presentation is probably more readable, but in our opinion tabular presentations have some advantages, e.g. they provide a more detailed insight into experimental data (reading values from a graph is less accurate).

Words removed from the results:

Page: 5  lines: 188; 202; 217; 264

  1. The legends on the histograms are redundant (too many legend panels of the same type) and could contain information on concentrations rather that letter acronyms, which are not intutive and provide no clarity.

Due to the fact that different concentrations are used for each mycotoxin (presented in Table 1, in accordance with literature data), the use of letter acronyms makes the interpretation of the results much easier. However, as suggested by the Reviewer, a single legend has been used for all data presented in Fig. 2.

  1. Since the repsonses are normalized to the control (i.e. 1), it's possible that including that bar in the plot is also redundant. All of these comments would help reduce the amount of physical and mind-space that the reader needs to expend to comprehend the material in the paper.

The bar corresponding to the control samples in Fig. 2 has been removed.

  1. To my mind, the conclusions of the paper are not very clear, while the discussion is overly long with at least some statements that are too speculative to be conclusions, since no data is presented to support them. Such as the statement in line 317-318 regarding the cell cycle. This statement could have been checked via monitoring of the cell cycle state of the yeast.

As suggested by both Reviewers, we have removed from the Discussion and Conclusions all statements that were not directly supported by our results and were only a discussion of the current state of knowledge. The changes introduced are marked in the manuscript.

Sentences removed from the discussion:

Page: 8  lines: 266-277; 283-287

Page: 9  lines: 305-307; 315-319; 324-326

Page: 10  lines: 351-354; 378-379; 391-394

Sentences removed from the conclusion:

Page:  15  lines: 621-622

  1. Lines 322-324 These statements could also easily be checked directly by monitoring the effects of antioxidant addition on biomass or marker induction

We are grateful for the Reviewer's valuable suggestion. The aim of our work was only to analyze the reaction of yeast to mycotoxins, therefore we did not investigate the effect of antioxidants on biomass or the induction of oxidative stress markers. At this stage of the research, there were no plans to introduce antioxidants into the media, although this is undoubtedly an interesting topic for further research.

  1. Lastly, non the described responses have been tested for being relevant for the toxic effects of the toxins. This could be done be searching the literature for whether mutations (deletions) of these proteins increased sensitivity to the toxins, or checking whether this was the case directly.

Since various biomarkers can be used to assess the stress occurring in aerobic cultures of microorganisms, the most adequate ones should be selected, allowing for the verification of the research hypothesis. The idea of simultaneously analyzing the entire range of available stress indicators in one publication seems questionable. However, we believe that the analysis of the activity of selected antioxidant enzymes in combination with the level of lipoperoxidation and the expression of heat shock proteins provides a satisfactory answer to the presented research problem.

  1. Overall, I think that the presented body of work is unsufficiently complete for a publication in IJMS, but, with some improvement, might be acceptable in a more specialized journal.

We do not avoid substantive polemics with the Reviewers, but the way this comment is formulated makes any discussion very difficult. We would be happy to provide explanations regarding the importance and relevance of the research topic undertaken, how the experiment was planned, the adequacy of the methods and techniques used, and the manner of presenting and discussing the results. Finally, we want to emphasize once again that the manuscript was submitted to the Special Issue of "Stress Response Research: Yeast as Models" after the Guest Editors confirmed that the topic of our study would be consistent with the profile of the journal and the special issue.

Reviewer 3 Report

Comments and Suggestions for Authors

Klosowski et al investigated  the effects of aflatoxin, ochratoxin and zeralenone on the growth of the commercial yeast strain Ethanol Red.
The the data reported in this manuscript indicate that mycotoxins  decrease yeast Biomass formation, although not proportionally to the concentration of mycotoxins added to the growth medium used during the experiments.
The authors also have investigated whether mycotoxin generate cellular oxidative stress by measuring the concentration of MDA, and the anzymatic activities of SOD, GPx and GST.
Furthermore, an increase in some Heat Shock Proteins was observed in yeast cells treated with mycotoxins.

The manuscript is well written and results are clearly reported.
In my opinion the manuscript should be accepetd in current form.

Round 2

Reviewer 1 Report

Comments and Suggestions for Authors

I accept the author's corrections.